# Effect of Age and Sex on the Quality of Offal and Meat of the Wild Boar (*Sus scrofa*)

**DOI:** 10.3390/ani10040660

**Published:** 2020-04-10

**Authors:** Agnieszka Ludwiczak, Joanna Składanowska-Baryza, Marek Stanisz

**Affiliations:** Department of Animal Breeding and Product Quality Assessment, Poznań University of Life Sciences, Słoneczna 1, 62-002 Suchy Las, Poland; joanna.skladanowska-baryza@up.poznan.pl (J.S.-B.); marek.stanisz@up.poznan.pl (M.S.)

**Keywords:** wild boar offal, wild boar meat, offal quality

## Abstract

**Simple Summary:**

Animal offal has been used for centuries in human nutrition as a source of valuable protein, vitamins and minerals. The goal of the study was to examine the effect of age and sex on the quality of offal and meat from the wild boar. A number of 32 hunt-harvested animals was assigned to groups according to age and sex. The quality of offal (liver, kidneys, heart and tongue) and meat (*m. semimembranosus*) of wild boars was examined. The analysed internal organs differed with their pH value. The meat in the group of sub-adults characterized with greater quality compared to the meat of juveniles i.a. better water holding capacity. The chemical composition of offal and meat from juveniles and sub-adults differed. The results of this study show that the quality attributes of offal and meat from the wild boar are affected by the animal’s age. This may suggest that different conditions/methods should be used in processing of these animal products to prevent spoilage (high pH shortens the shelf life), the loss of residual water, and offer consumers products with acceptable color.

**Abstract:**

The goal of the study was to examine the effect of age and sex on the quality of wild boar offal and meat. A number of 32 hunt-harvested animals was assigned to groups according to age (juveniles and sub-adults) and sex. The quality of offal (liver, kidneys, heart and tongue) and *m. semimembranosus* was examined. The pH value of *m. semimembranosus* ranged from 5.45 to 5.88. The highest pH was recorded in the kidney and the liver (6.32–6.54 and 6.12–6.31). The meat in the group of juveniles was brighter (*p* = 0.042), yellower (*p* = 0.039), showed a greater drip loss (*p* = 0.007), cooking loss (*p* = 0.039), and plasticity (*p* = 0.028), compared to the sub-adults. The extractable fat content in the *m. semimembranosus* and offal (*p* = 0.004), and water to crude protein ratio (*p* = 0.033), also differed between age groups. The results of the study show different quality attributes of offal and meat of wild boars from two age groups. The obtained quality measures suggest that the culinary and technological usefulness of offal and meat from the wild boars may differ according to the age of hunted animals.

## 1. Introduction

The most bothering factors connected with acquisition of game species are related to the biosecurity and safety of game meat consumption [1]. But the fact is that, despite its disadvantages, the meat of game species, like deer or wild boars, is popular among some groups of consumers. Meat is known to be one of the most expensive sources of protein in human nutrition. Offal is also nutritionally valuable and cheaper at the same time, but sometimes considered as less noble compared to meat. Offal is the name of non-carcass parts, also referred to as by products [2]. Many studies on the quality of offal confirm their usability for human nutrition, as a good source of valuable protein, vitamins and minerals [3,4,5]. According to scientific studies dealing with game meat, this animal product may be negatively affected by the hunting method or improperly conducted postmortem procedures related with dressing and chilling the carcasses [6,7,8]. Despite the mentioned problems, game meat has benefits over meat from domestic animals. The wild boar meat is known to be tasty and nutritionally valuable [9,10]. Moreover Poland is dealing with an overpopulation of the wild boar. According to the Polish Central Statistical Office [11,12] the population of the wild boar increased from 310 thousand (the hunting season 2016/2017) to 341 thousand (2017/2018) in recent years. The authors of studies conducted on wild boars hunt-harvested in Poland in the end of XX century have correctly predict, that the age structure of the population changes, and most of the obtained animals will characterize with the body weight below 50 kg [13,14]. This fact has dictated the age classes distinguished in the presented study. The novelty of this study is the physicochemical analysis of wild boars’ offal, including traits deciding about the shelf life and consumer acceptance of meat and meat products. The goal was to analyze the effect of age and sex on the quality of wild boar meat and offal.

## 2. Materials and Methods

The 32 wild boars analysed in the present study were shot in the Northwest Poland, in January. The animals were shot during a regular group hunt conducted according to the Polish hunting law, therefore the study did not require an approval of Ethical Committee (the study material was obtained after the animals were hunt-harvested). The age was estimated on the basis of tooth wear and replacement, and the animals were assigned to groups according to their age and sex: 18 juveniles (<1 year; 10 males and 8 females) and 14 sub-adults (1–2 years; 7 males and 7 females). After the shot the wild boars were eviscerated in the hunting area and at the same time the offal were obtained (liver, kidneys, heart and tongue), and kept under chilled conditions until the analysis. Then the animals were transported from the hunting area to the game establishment were they went through veterinary inspections, were weighed together with the head and skin (to obtain the dressed field weight), dressed and placed in the chilling room (+2 °C). The offal was also weighted after the veterinary inspection. The muscles (*m. semimembranosus*) were obtained from the carcasses 24 h post mortem. All the samples (muscles and offal) were stored under chilled conditions (+2 °C) until the analysis. All the analysis were repeated twice per a sample.

The pH of the muscle and offal was examined 24 h and 48 h postmortem. The measures were taken with a temperature compensated, combination glass calomel electrode (ERH-11X1, SCHOTT, Mainz, Germany) connected to a portable pH-meter (Handylab 2, SCHOTT, Mainz, Germany). Before the measurements the pH equipment was calibrated in pH 4.0 and 7.0 buffers.

The colour parameters of *m. semimembranosus* samples were measured 24 h and 48 h postmortem with the Minolta colorimeter CR-200b (illuminant D65, 2° observer with a 8-mm-diameter aperture size; Konica Minolta, The Netherlands). The meat samples were allowed to bloom for 45 min at +4 °C before measurement. The CIE system was used for the measurement of lightness (L*), redness (a*), and yellowness (b*) [15]. The chroma (C*) and hue-angle (h°) was calculated automatically by the Spectra Magic program of the CR-200b instrument.

The drip loss, cooking loss, total water, free water, and plasticity of the *m. semimembranosus* were measured 24 h postmortem:

The drip loss (%) and cooking loss (%) were measured according to Honikel [16]. The muscle slices (2.5-cm-thick and 40–50 g) were cut perpendicular to the direction of the muscle fibers. The samples for the measurement of drip loss were weighed, hung on hooks, and placed in a container to reduce evaporation (+2 °C). After 24 h, the samples were reweighed to calculate the change in their weight. The muscle slices for the measurement of cooking loss were wrapped in thin plastic bags. The bags with meat were placed in a water bath set at 90 °C until reaching the core temperature of 70 °C (measured with thermocouples). Then, the samples were cooled to room temperature and reweighed (after removing excess moisture with a paper towel). Changes in the sample weight were calculated (%).

The free water (%) was measured using a filter-paper press method, according to Grau and Hamm [17] as modified by Pohja and Niinivaara [18]. Samples (0.300 g) of ground *m. semimembranosus* were placed on a filter paper between two glass tiles. A force of 2 kg was applied on each sample for 5 min. Then, samples were removed from the filter paper and immediately reweighed, to calculate the change in their weight.

Plasticity (cm^2^) measurement was conducted according to Pohja and Niinivaara [18], simultaneously to the free water measurement. The meat plasticity was expressed as the area of the compressed meat sample used for the measurement of free water.

The analyses of the chemical composition [19] of wild boar muscle and offal were made 24 h postmortem and included: the determination of dry matter content (the samples were dried at 105 °C to a constant weight), the determination of the total protein content with the Kjeldahl procedure (K-424 Buchi digestion unit; Büchi Labortechnik AG, Switzerland), and the determination of extracted fat content using Soxhlet extraction with diethyl ether (MLL 147, AJL Electronics, Cracow, Poland).

The dataset was checked for normality and followed normal distribution. The effect of age and sex on the dressed field weight, and the weight and percentage of offal (liver, kidneys, heart, tongue) in the dressed field weight, was calculated by means of the two-way ANOVA.

The effect of age, sex, the point of measurement (*m. semimembranosus*, liver, kidneys, heart, tongue) and time postmortem on the pH was calculated by means of the nested model. The point of measurement was nested in the animal, and the time postmortem was nested in the point of measurement as repeated measures of pH.

The effect of age, sex, and the point of measurement (*m. semimembranosus*, liver, kidneys, heart, tongue) on the dry matter, crude protein, extractable fat, water/crude protein was calculated with a nested model, where the point of measurement was nested in the animal.

The effect of age, sex and time postmortem on the muscle color (L*, a*, b*, C*, H°) was calculated with the nested model, in which the time postmortem was nested in the muscle as repeated measures of color.

Main effects with a *p*-Value ≤ 0.05 were reported as significant. There were no interactions between the analysed effects, therefore they were not included in the models. The dressed field weight was included in the models as a covariate. Tukey–Kramer adjustment was implemented for multiple comparisons of Least Squares (LS) mean differences. All the statistical analyses were made with SAS ver. 9.4 software package (SAS Inst. Inc., Cary, NC, USA) [20].

## 3. Results

The effect of age and sex on the dressed field weight of wild boars, and the weight and proportion of offal, was analysed in the present study (Table 1).

No difference in the dressed field weight was observed between the juvenile males and females. The effect of sex was noted among older animals, with the males having greater dressed field weight than females (*p* = 0.047). Obviously, the older animals were heavier compared to the younger ones (*p* = 0.001). The proportion of internal organs (liver, kidneys and heart) in the dressed field weight of juveniles was greater compared to the sub-adults. Among the sub-adults, the females characterised with lower weight of kidneys (*p* = 0.043) and heart (*p* = 0.024) compare to males.

The differences in the pH values of the analysed points of measurement according to age, sex and the time postmortem are presented in Table 2, while the P-values are given in Table 3. The pH of *m. semimembranosus* ranged from 5.45 to 5.88. Among the examined offal, the highest pH values were found for the kidneys and the liver (6.32–6.54 and 6.12–6.31). No effect of age, sex, time postmortem and the dressed field weight on the pH of analyzed points of measurement was noted (Table 3).

The colour parameters of *m. semimembranosus* are presented in Table 4. The meat brightness, yellowness and hue angle increased during the chilled storage (*p* = 0.019; *p* = 0.001 and *p* = 0.001). The meat from juveniles characterised with a slightly higher L* and b*, compared to the meat from sub-adults (*p* = 0.042 and *p* = 0.039). The capacity of *m. semimembranosus* to hold residual water and the content of water fractions were also examined (Table 5). The drip loss (*p* = 0.007), cooking loss (*p* = 0.039) and plasticity (*p* = 0.028) were greater in the meat from juveniles compared to the meat of sub-adults. A slight difference in the content of free water was found between meat from males and females (*p* = 0.018). The meat of males also characterised with a greater drip loss compared to the meat of females (*p* = 0.039).

The differences in the chemical composition of the analysed points of measurement according to age and sex of the wild boars are presented in Table 6, while the *p*-values are given in Table 7. The extractable fat content was greater in the *m. semimembranosus* and offal from sub-adults, when compared to the composition of organs from younger animals (*p* = 0.004). The water to crude protein ratio of meat from two age groups was significantly different (*p* = 0.033).

## 4. Discussion

The effect of age on the body composition is related to the sequential development of muscle, bone and fat tissue [21]. The dressed field weights presented in our study fit in the range of weights of wild boars hunted in Europe: youngs (7–12 months) weight from 24.6 kg (in Switzerland) to 30.0 kg (in Czech Republic), while sub-adults (13–24 months) weight from 37.5 kg (in Poland) to 64.9 kg (in Czech Republic) [22]. On the contrary, Drozd et al. [23] found different dressed field weights of wild boars hunted in Poland: juveniles weighted 29.0 kg, yearlings weighted 48.9 kg, while the adults weighted 83.9 kg. Babicz et al. [24] studied the proportion of internal organs in the dressed filed weight of wild boars (51.9–54.3 kg). These authors found a higher proportion of heart (0.79%) and kidneys (0.57%), and lower proportion of tongue (0.43%) in the body weight of wild boars, compared to our study. Unlike domestic pigs, wild boars are characterised with a greater proportion of by products in their body weight. In their study, Babicz et al. [24] found that the proportion of tongue, heart, liver and kidneys in the body of 113.6–115.3 kg fatteners corresponds to 0.26%, 0.37%, 1.41% and 0.15%. On the other hand, Migdał et al. [25] presented an even higher proportion of internal organs in the body weight of 110 kg fatteners, measuring 0.36–0.41% for the heart, 2.04–2.17% for the liver and 0.36–0.43% for the kidneys. According to the study conducted by Razmaite et al. [26], an effect of crossing domestic pigs (Lithuanian indigenous wattle pigs) with wild boars can be observed in the weight of internal organs. The heart was much heavier in crossbreds with ½ proportion of the wild boar genotype (0.38 kg) compared to the pure domestic breed (0.28 kg) and the crossbreds with ¼ wild boar genotype (0.30 kg). The liver was heavier in the Lithuanian pigs (1.53 kg) compared to ½ proportion of the wild boar genotype (1.41 kg). The results in our study show an expected effect of age and sexual dimorphism on the slaughter traits of the examined animals, while the dissimilarity with results of other research are connected with the effect of genotype and/or environment.

The liver, kidneys, heart and tongue are structurally different organs, with a contrasting scheme of metabolic activity. The pH value of meat of the wild boar varies considerably from 5.4 to 6.29 [6,10,27,28]. In the present study, the pH value of *m. semimembranosus* measured even >5.8 in the group of juvenile females, stipulating an intermediate DFD (dark, firm and exudative) [10]. If so, it should be accompanied by dark colour and low water holding capacity, and none of those observations were made. In our research, similar pH values were found for the kidneys and the liver, resulting from their structural similarity (glandular structure). These organs also showed the highest level of pH. Similar observations were made by other authors. According to study conducted by Babicz et al. [24], the pH of wild boar internal organs (24 h postmortem) measures 5.86 in the tongue, 5.79 in the heart, 6.13 in liver, and 6.40 in kidneys. In the study of Kropiwiec et al. [29], the pH in the internal organs of two genetic group of fatteners measured 5.68 and 5.83 in the tongue, 5.71 and 5.83 in the heart, 5.61 and 5.89 in the liver, 5.97 and 6.10 in the kidneys. Tomović et a. [30] examined the physicochemical traits of internal organs obtained from males Swallow-Belly Mangalica kept in the free-range system. The authors obtained a following pH values: 5.74 in the tongue, 5.85 in the heart, 6.07 in the liver and 6.47 in the kidneys. On the basis of the available literature on the quality of internal organs of wild boars and domestic pigs, we can consider the pH values found in the presented study as optimal.

A slight effect of age on L* of wild boar *m. semimembranosus* was also found in the study conducted by Stanisz et al. [10]. The juveniles characterized with a brighter meat (L* ranging from 41.8 to 43.5) compared to yearlings (L* ranging from 38.7 to 40.3). The following researchers: Pedrazzoli et al. [28], Marchiori and Felício [31], Florek et al. [32] and Kasprzyk et al. [33], noted a great variability in range of wild boar meat colour parameters, most probably connected with the age, muscle type, diet as well as the level of bleeding out after the shot. According to Kasprzyk et al. [33], the colour of meat from wild boars is affected by the carcass weight. They observed that brighter meat is typical for lighter animals (carcass weight of 30 kg and 45 kg), while wild boars with a carcass weight ≥60 kg characterize with darker meat.

In the presented study, the higher drip loss and cooking loss were found in the meat from juveniles compared to sub-adults. A conclusion may be drawn that the meat from sub-adults is more suitable for chilled storage and thermal processing compared to the meat from juveniles, because lower water losses can be expected. Compared to our study, Babicz et al. [34] presented lower values of free water percentage, measuring 22.35–22.89% in the *m. longissimus lumborum*, and 22.10–22.35% in the *m. adductor femoris* from wild boar crosses with domestic breeds. While Kasprzyk et al. [33] obtained the free water measurement in a range of 20.10–24.48% in the LTL, and 20.74–23.09% in the *m. semimembranosus.* The cooking loss values obtained in our study were optimal for the meat of the wild boar [6,10]. Borilova et al. [27] in their study, give even higher cooking losses, measuring 36.74% in the meat from the shoulder and 37.08% in the meat from the leg. In the study of Kasprzyk et al. [33] the cooking loss measured from 32.01 to 35.71% in the LTL muscle, and from 33.53 to 36.71% in the *m. semimembransus*. The measures of drip loss, free water and cooking loss are highly affected by the factors like the used method and muscle type, which are probably the major source of variability among the mentioned meat quality traits.

The proximal chemical composition of wild boar’s offal may show a great variety. The effect of age and sex on the chemical composition of meat (*m. longissimus*) from the wild boar was found by Dannenberg et al. [35]. The authors observed a higher content of protein in meat from males compared to females, and no differences in the content of extractable fat and water. Tesarova et al. [36] found that the highest content of protein characterised the *m. teres major* of females aged 13–24 months, compared to females ages 1–12 months. In their study, the meat of younger females characterised with the highest content of extractable fat measuring 2.83%. Babicz et al. [24] analysed the chemical composition of the tongue, heart, liver and kidneys of wild boars. The authors obtained 14.17%, 3.05%, 7.82% and 4.84% of extractable fat, and 15.86%, 20.54%, 29.86%, 20.19% of crude protein in the mentioned organs. Comparing the results presented by Babicz et al. [24] with the data from our study, significant differences can be found in the content of CP in the liver, EF in the heart, CP and EF in kidneys. Similarly to the observation from our study, Tomović et al. [30] presented the tongue of the Swallow-Belly Mangalica contains much more extractable fat (204 g kg^−1^) compared to the heart (73.2 g kg^−1^), liver (31.0 g kg^−1^) and kidney (37.9 g kg^−1^). The same authors pointed that the liver is better source of protein compared to the tongue, heart, liver and kidneys (188.5 g kg^−1^ vs. 150.7 g kg^−1^, 157.0 g kg^−1^_,_ and 147.9 g kg^−1^). The heart and the kidney characterised with the greatest content of total water (759.4 g kg^−1^ and 801.5 g kg^−1^). The available research underline that the animal offal are a valuable food, and may be used as a cheap source of protein in human diet, though in some parts of the world they are considered as less noble food.

## 5. Conclusions

Our studies point out the good qualities of offal and meat of the wild boar male juveniles, and male and female sub-adults. The meat of juvenile females can be categorized as medium quality meat, because of the increased pH value. Some of the meat quality attributes (colour, water compartments and capacity to hold residual water) and the proximal chemical composition (content of extractable fat and water to crude protein ratio) of analysed offal and meat varied among age and sex groups. This may suggest that different conditions/methods should be applied in processing of these animal products to prevent spoilage (high pH shortens the shelf life), the loss of residual water, and offer consumers products with acceptable colour. On the basis of the obtained results conclusions may be drawn about different culinary and technological usefulness of offal and meat from the wild boars, according to the age of hunted animals.

## Figures and Tables

**Table 1 animals-10-00660-t001:** The dressed field weight, mass and proportion of edible offal of the wild boar (mean ± SD).

Item		Age				Effect (*p*-Value)	
		Juveniles		Sub-Adults		Age	Sex
		**Male**	**Female**	**Male**	**Female**		
DFW	kg	26.7 ± 1.1 ^A^	24.3 ± 1.1 ^A^	60.4 ± 1.6 ^Ba^	55.9 ± 1.6 ^Bb^	0.001	0.047
Liver	g	637 ± 23 ^A^	634 ± 22 ^A^	1168 ± 29 ^B^	1108 ± 29 ^B^	0.001	0.279
	% ^1^	2.52 ± 0.08 ^A^	2.55 ± 0.08 ^A^	1.87 ± 0.09 ^B^	1.98 ± 0.09 ^B^	0.001	0.728
Kidneys	g	126 ± 7 ^A^	119 ± 7 ^A^	237 ± 8 ^Ba^	196 ± 9 ^Bb^	0.001	0.043
	% ^1^	0.49 ± 0.03 ^a^	0.49 ± 0.03 ^a^	0.40 ± 0.03 ^b^	0.37 ± 0.03 ^b^	0.008	0.829
Heart	g	185 ± 14 ^A^	176 ± 16 ^A^	387 ± 14 ^Ba^	320 ± 16 ^Bb^	0.001	0.024
	% ^1^	0.69 ± 0.02 ^A^	0.68 ± 0.02 ^A^	0.60 ± 0.02 ^B^	0.57 ± 0.02 ^B^	0.001	0.248
Tongue	g	147 ± 11 ^A^	135 ± 11 ^A^	303 ± 12 ^B^	275 ± 12 ^B^	0.001	0.093
	% ^1^	0.56 ± 0.03	0.53 ± 0.03	0.50 ± 0.03	0.49 ± 0.03	0.058	0.359

^1^ % of the dressed field weight (together with the head and skin); Means within the same row marked with superscripts A, B (a, b) differ significantly at *p* < 0.01 (*p* < 0.05); DFW—dressed field weight.

**Table 2 animals-10-00660-t002:** The pH values of meat and edible offal from the wild boar (mean ± SD).

PM	Age							
	Juveniles				Sub-Adults			
	Male		Female		Male		Female	
	**24 h**	**48 h**	**24 h**	**48 h**	**24 h**	**48 h**	**24 h**	**48 h**
MS	5.69 ± 0.09 ^A^	5.78 ± 0.09 ^A^	5.82 ± 0.08 ^A^	5.88 ± 0.08 ^AC^	5.45 ± 0.11 ^A^	5.49 ± 0.11 ^A^	5.58 ± 0.11 ^A^	5.62 ± 0.11 ^A^
Liver	6.26 ± 0.09 ^B^	6.29 ± 0.09 ^B^	6.27 ± 0.08 ^B^	6.31 ± 0.08 ^B^	6.13 ± 0.11 ^B^	6.17 ± 0.11 ^B^	6.12 ± 0.11 ^BC^	6.14 ± 0.11 ^BC^
Kidneys	6.44 ± 0.09 ^B^	6.45 ± 0.09 ^B^	6.39 ± 0.08 ^B^	6.45 ± 0.08 ^B^	6.41 ± 0.12 ^C^	6.54 ± 0.12 ^C^	6.32 ± 0.12 ^B^	6.34 ± 0.12 ^B^
Heart	5.91 ± 0.08 ^C^	5.89 ± 0.08 ^C^	5.96 ± 0.08 ^A^	5.86 ± 0.08 ^A^	5.81 ± 0.12 ^D^	5.85 ± 0.12 ^D^	5.92 ± 0.12 ^C^	5.98 ± 0.12 ^C^
Tongue	5.52 ± 0.08 ^A^	5.58 ± 0.08 ^A^	5.57 ± 0.08 ^C^	5.53 ± 0.08 ^C^	5.61 ± 0.12 ^AD^	5.66 ± 0.12 ^AD^	5.55 ± 0.12 ^AD^	5.59 ± 0.12 ^AD^

Means within the same column marked with superscripts A, B, C, D are differ significantly at *p* < 0.01; PM—point of measurement; MS—*m. semimembranosus.*

**Table 3 animals-10-00660-t003:** The *p*-Values for the effect of age, sex, point of measurement, time postmortem and the dressed field weight on the pH of meat and offal.

Trait	Effect (*p*-Value)				
	Age	Sex	PM	TP	DFW
pH	0.462	0.537	<0.0001	0.132	0.563

PM—point of measurement; TP—time postmortem; DFW—dressed field weight.

**Table 4 animals-10-00660-t004:** The colour parameters of *m. semimembranosus* from the wild boar (mean ± SD).

Trait	Age								Effect (*p*-Value)			
	Juveniles				Sub-Adults							
	Male		Female		Male		Female					
	24 h	48 h	24 h	48 h	24 h	48 h	24 h	48 h	Age	Sex	TP	DFW
L*	41.9 ± 0.9 ^ab^	43.1 ± 0.9 ^a^	41.6 ± 0.9 ^ab^	43.8 ± 0.9 ^a^	38.8 ± 0.9 ^b^	41.8 ± 0.9 ^a^	39.1 ± 0.9 ^b^	42.1 ± 0.9 ^a^	0.042	0.179	0.019	0.728
a*	11.3 ± 0.7	11.2 ± 0.7	11.7 ± 0.7	11.3 ± 0.7	12.9 ± 0.9	12.4 ± 0.9	12.6 ± 0.8	12.3 ± 0.8	0.724	0.236	0.659	0.302
b*	6.1 ± 0.7 ^a^	7.1 ± 0.7 ^ab^	6.4 ± 0.5 ^ab^	7.7 ± 0.5 ^b^	5.5 ± 0.7 ^a^	5.6 ± 0.7 ^a^	6.2 ± 0.7 ^a^	7.7 ± 0.7 ^b^	0.039	0.114	0.001	0.714
C*	12.8 ± 0.8	13.3 ± 0.8	13.4 ± 0.8	13.9 ± 0.8	14.1 ± 0.8	14.9 ± 0.8	14.0 ± 0.8	14.6 ± 0.8	0.951	0.219	0.318	0.371
h°	28.1 ± 2.7 ^AB^	32.3 ± 2.7 ^B^	28.3±1.9 ^AB^	34.4 ± 1.9 ^B^	23.3 ± 2.7 ^A^	31.4 ± 2.7 ^B^	26.1 ± 1.9 ^A^	32.1 ± 1.9 ^B^	0.523	0.589	0.001	0.801

Means within the same row marked with superscripts A, B (a, b) are differ significantly at *p* < 0.01 (*p* < 0.05); TP—time postmortem; DFW—dressed field weight; L*—lightness; a*—redness; b*—yellowness; C*—chroma; h°—hue.

**Table 5 animals-10-00660-t005:** The capacity to hold residual water, water fractions and plasticity of *m. semimembranosus* from the wild boar (mean ± SD).

Trait	Age				Effect (*p*-Value)		
	Juveniles		Sub-Adults				
	Male	Female	Male	Female	Age	Sex	DFW
Drip loss (%)	3.34 ± 0.28 ^a^	3.12 ± 0.27 ^a^	2.71 ± 0.27 ^ab^	2.03 ± 0.26 ^c^	0.007	0.039	0.898
Total water (%)	75,96 ± 0.54	76.64 ± 0.60	75.84 ± 0.61	74.96 ± 0.59	0.363	0.626	0.969
Free water (%)	30.14 ± 0.98 ^AB^	30.28 ± 0.98 ^AB^	31.73 ± 0.98 ^A^	28.15 ± 0.55 ^B^	0.879	0.018	0.339
Cooking loss (%)	33.32 ± 0.85	33.38 ± 0.85	30.77 ± 0.85	30.76 ± 0.85	0.039	0.983	0.315
Plasticity (cm^2^)	4.21 ± 0.38 ^a^	4.43 ± 0.36 ^a^	3.48 ± 0.36 ^b^	3.47 ± 0.35 ^b^	0.028	0.676	0.226

Means within the same row marked with superscripts A, B (a, b, c) are differ significantly at *p* < 0.01 (*p* < 0.05); DFW—dressed field weight.

**Table 6 animals-10-00660-t006:** The proximal chemical composition of muscle and offal from the wild boar (mean ± SD).

Organs	Age															
	Juveniles								Sub-Adults							
	Male				Female				Male				Female			
	DM	CP	EF	W/CP	DM	CP	EF	W/CP	DM	CP	EF	W/CP	DM	CP	EF	W/CP
MS	24.04 ± 0.67 ^A^	21.83 ± 0.47 ^Aa^	1.34 ± 0.28 ^AC^	3.49 ± 0.12 ^ACa^	23.36 ± 0.67 ^Aa^	22.12 ± 0.47 ^A^	1.47 ± 0.28 ^AB^	3.48 ± 0.12 ^A^	24.16 ± 0.67 ^A^	22.81 ± 0.47 ^A^	1.63 ± 0.28 ^AB^	3.34 ± 0.12 ^A^	24.04 ± 0.67 ^A^	22.83 ± 0.47 ^Aa^	1.87 ± 0.28 ^A^	3.28 ± 0.11 ^A^
Liver	29.44 ± 0.73 ^B^	20.83 ± 0.52 ^Ab^	1.94 ± 0.29 ^A^	3.39 ± 0.14 ^A^	29.45 ± 0.73 ^B^	21.38 ± 0.52 ^A^	2.01 ± 0.29 ^A^	3.16 ± 0.14 ^A^	28.08 ± 0.73 ^B^	21.49 ± 0.52 ^B^	2.09 ± 0.29 ^A^	3.36 ± 0.14 ^A^	29.59 ± 0.73 ^B^	21.72 ± 0.52 ^Ab^	2.17 ± 0.29 ^Aa^	3.24 ± 0.11 ^A^
Kidneys	21.05 ± 0.66 ^C^	16.79 ± 0.47 ^Ba^	1.02 ± 0.29 ^BC^	4.75 ± 0.14 ^Ba^	21.20 ± 0.66 ^Ab^	17.33 ± 0.47 ^B^	1.12 ± 0.29 ^B^	4.57 ± 0.14 ^BCa^	22.15 ± 0.66 ^C^	16.54 ± 0.47 ^C^	1.23 ± 0.29 ^B^	4.74 ± 0.14 ^B^	21.71 ± 0.66 ^C^	17.01 ± 0.47 ^B^	1.53 ± 0.29 ^Ab^	4.62 ± 0.11 ^B^
Heart	24.06 ± 0.66 ^A^	19.23 ± 0.45 ^C^	1.46 ± 0.29 ^AC^	3.95 ± 0.13 ^CDb^	23.51 ± 0.66 ^Aa^	19.01 ± 0.45 ^C^	1.48 ± 0.29 ^AB^	4.03 ± 0.13 ^Bb^	24.47 ± 0.66 ^A^	19.24 ± 0.45 ^B^	1.52 ± 0.29 ^AB^	3.92 ± 0.13 ^C^	23.01 ± 0.66 ^AC^	18.87 ± 0.45 ^C^	1.79 ± 0.29 ^A^	4.08 ± 0.11 ^C^
Tongue	31.62 ± 0.86 ^D^	15.72 ± 0.58 ^Bc^	12.58 ± 0.34 ^D^	4.36 ± 0.13 ^BDc^	30.96 ± 0.86 ^C^	14.77 ± 0.58 ^D^	12.54 ± 0.34 ^C^	4.68 ± 0.13 ^DC^	32.93 ± 0.86 ^D^	16.17 ± 0.58 ^C^	13.92 ± 0.34 ^C^	4.17 ± 0.13 ^C^	32.96 ± 0.86 ^D^	17.15 ± 0.58 ^B^	14.85 ± 0.34 ^B^	3.92 ± 0.11 ^C^

Means within the same column marked with superscripts A, B, C, D (a, b, c) are differ significantly at *p* < 0.01 (*p* < 0.05); MS—*m. semimembranosus*; DM—dry matter, CP—crude protein, EF—extractable fat, W/CP—water/crude protein ratio.

**Table 7 animals-10-00660-t007:** The *p*-Values for the effect of age, sex, point of measurement and carcass weight on the proximal chemical composition of meat and offal from the wild boar.

Trait	Effect (*p*-Value)			
	Age	Sex	PM	DFW
DM	0.363	0.626	0.001	0.969
CP	0.143	0.248	0.001	0.658
EF	0.004	0.048	0.001	0.943
W/CP	0.033	0.437	0.001	0.461

DM—dry matter, CP—crude protein, EF—extractable fat, W/CP—water/crude protein ratio, PM—point of measurement; DFW—dressed field weight.

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
