# Peer review of "Effect of Age and Sex on the Quality of Offal and Meat of the Wild Boar (Sus scrofa)"

_animals, 2020, doi:10.3390/ani10040660_

Round 1

Reviewer 1 Report

Observatios in attachment.

Author Response

Reviewer 1. The research is of scientific interest and relevant from the point of view of meat technology of wild animals

Authors reply: We thank the reviewer for the valuable comments and suggestions. We modified the text, following the reviewer’s indications. We address point by point the reviewer’s comments below. All the changes in the text and tables were marked.

Materials and methods

Line 62: “The age has been estimated” change to “The change was estimated”. Past tense.

Corrected

Line 63: Animals were assigned instead “animals have been assigned”.

Corrected

Lines 65-66: Were the visceras maintened in refrigeration during its transportation to the game establishment?

The viscera were kept under chilled conditions. Added in line 66.

Lines 72, 76, 83: Eliminate space: post mortem

Corrected throughout the manuscript.

Line 106: dressed carcass weight or dressed field weight?. In the discussion section is called Dressed carcass weight. Uniformize in all the manuscript.

Corrected to dressed field weight throughout the manuscript.

Line 119: The effect of dressed field weight was included as partial correlation coefficient or covariate in the model?

We analysed the effect of the dressed field weight as covariate. Corrected in lines 121-122.

Statistic Analysis section: Indicate the p-value to estimate statistical significances.

Main effects with a P-value ≤ 0.05 were reported as significant – placed in line 120.

Results

Line 127: was observed instead has been observed.

Corrected

Table 2. If only Muscle /offal effect was observed, then the legend in the footnote (Table 2), should indicate that the statistical differences are between means within the column and not within the row.

Corrected

Table 6. Same comment that table 2. According to the statistical significance shown in Table 7, some superindices were assigned by rows and others within columns. The interpretation of these results is confusing. Correct the legend of the footnote in table 6.

The numbers of Tables were changed. Previous Table 3, was changed to Table 2a, as the P-values refer to the means given in Table 2. Previous Table 7 became Table 5a, as the P-values in this table refer to Table 5. A proper explanation was placed in the Results section, in lines 136-137, 154-155.

The legends in tables were corrected.

Reviewer 2 Report

As we know, the consumer acceptance of meat of wild animals, especially edible offal parts, is quite diverse. For some consumers, this is a delicacy of a high gastronomic and nutritional value, for some, as the Authors stated, this is less novel food and this is due to many factors such as organoleptic characteristics, safety and hygiene issues, high price or difficult access. However, the population of wild boars is raising, so the studies concerning the quality of their meat are needed, and in recent years a few similar studies were published. In my opinion, a sensory evaluation would be a great addition to the study, especially, since the Authors highlight that the novelty of the study is assessing the relationship between the physicochemical traits and consumer acceptance.

Question/Remarks:

  • Please correct throughout the entire manuscript, physiochemical to physicochemical properties - this is a common mistake made in the publications.
  • How many measurements were made per sample?
  • Did the Authors check the data for normality and homogeneity? There is no information in the work. 
  • In my opinion, there is no point to include the muscle/offal in the statistical model. Age, sex and time of measurements as experimental factors might have an impact. But, the difference between muscle and organs is obvious and comparison of such samples as m. semimembranosus and kidney, for example, has no point. The Authors notice this themselves (lines 191-192). Thus, I suggest excluding this parameter from the statistical analysis.    

Some technical remarks:

-lines 56-57- "We did not focus on the examination of microbiological quality of wild boar meat and organs." - please delete

-line 38 - ..of game meat consumption.. - add "meat"

-lines 53-54 - "The novelty of this study is the assumption to
look at the wild boars’ offal from the perspective of .." - style, please correct

-m. semimembranosus - the Latin name is written once in italics once not (is. Table 4-5; Line 194 etc.), please unify it in the text;

-Table 5 - title: The capacity of to hold - style

-Table 5 - "TPM" was not analyzed as an experimental factor - please remove from the table description

-post-mortem or postmortem? please check in the entire text

-Table 6 - SD values in each subgroup were exactly the same? I know they can be close, but the same in each parameter in each group?

-Table 7 - the title of the table and the table itself are divided with "Discussion" section - please correct

-line 227 eta al. - please correct

Author Response

Reviewer 2. As we know, the consumer acceptance of meat of wild animals, especially edible offal parts, is quite diverse. For some consumers, this is a delicacy of a high gastronomic and nutritional value, for some, as the Authors stated, this is less novel food and this is due to many factors such as organoleptic characteristics, safety and hygiene issues, high price or difficult access. However, the population of wild boars is raising, so the studies concerning the quality of their meat are needed, and in recent years a few similar studies were published. In my opinion, a sensory evaluation would be a great addition to the study, especially, since the Authors highlight that the novelty of the study is assessing the relationship between the physicochemical traits and consumer acceptance.

Authors reply: We thank the reviewer for the valuable comments and suggestions. We modified the text, following the reviewer’s indications. We address point by point the reviewer’s comments below. All the changes in the manuscript were marked.

Question/Remarks:

  • Please correct throughout the entire manuscript, physiochemical to physicochemical properties - this is a common mistake made in the publications.

Corrected.

  • How many measurements were made per sample?

Added in line 72.

  • Did the Authors check the data for normality and homogeneity? There is no information in the work. 

The dataset was checked for normality and followed normal distribution – placed in line 107.

  • In my opinion, there is no point to include the muscle/offal in the statistical model. Age, sex and time of measurements as experimental factors might have an impact. But, the difference between muscle and organs is obvious and comparison of such samples as  semimembranosus and kidney, for example, has no point. The Authors notice this themselves (lines 191-192). Thus, I suggest excluding this parameter from the statistical analysis.    

The model given in the manuscript was consulted with a statistician. Because there quality measures are repeated in time, we were suggested to nest the muscle and organs in the animal. The appraisal of the tables results from the nested model.

We have changed the term ‘muscle/ organs’ to ‘point of measurement’ – maybe this form will be more acceptable and reflects the nested procedure more clearly.

Some technical remarks:

-lines 56-57- "We did not focus on the examination of microbiological quality of wild boar meat and organs." - please delete

Deleted

-line 38 - ..of game meat consumption.. - add "meat"

Added

-lines 53-54 - "The novelty of this study is the assumption to
look at the wild boars’ offal from the perspective of .." - style, please correct

Corrected to: The novelty of this study is the physicochemical analysis of wild boars’ offal, including traits, that are commonly used as indicators of shelf life and consumer acceptance in case of meat and meat products.

-m. semimembranosus - the Latin name is written once in italics once not (is. Table 4-5; Line 194 etc.), please unify it in the text;

Corrected throughout the text

-Table 5 - title: The capacity of to hold – style

Corrected

-Table 5 - "TPM" was not analyzed as an experimental factor - please remove from the table description

Removed

-post-mortem or postmortem? please check in the entire text

Corrected to post-mortem throughout the text.

-Table 6 - SD values in each subgroup were exactly the same? I know they can be close, but the same in each parameter in each group?

This similarity results from the fact, that the original SD values generated by the program were rounded down to two decimal places.

-Table 7 - the title of the table and the table itself are divided with "Discussion" section - please correct

Corrected

-line 227 eta al. - please correct

Corrected